# Determination of Constrained Modulus of Granular Soil from In Situ Tests—Part 1 Analyses

**K. Rainer Massarsch** [ORCID]

Geo Risk and Vibration Scandinavia AB, SE 168 41 Bromma, Sweden; rainer.massarsch@georisk.se

**Abstract:** Assessing the constrained modulus is a critical step in calculating settlements in granular soils. This paper describes a novel concept of how the constrained modulus can be derived from seismic tests. The advantages and limitations of seismic laboratory and field tests are addressed. Based on a comprehensive review of laboratory resonant column and torsional shear tests, the most important parameters affecting the shear modulus, such as shear strain and confining stress, are defined quantitatively. Also, Poisson's ratio, which is needed to convert shear modulus to constrained modulus, is strain-dependent. An empirical relationship is presented from which the variation in the secant shear modulus with shear strain can be defined numerically within a broad strain range ($10^{-4}$–$10^{-0.5}$%). The tangent shear modulus was obtained by differentiating the secant shear modulus. According to the tangent modulus concept, the tangent constrained modulus is governed by the modulus number, $m$, and the stress exponent, $j$. Laboratory test results on granular soils are reviewed, based on which it is possible to estimate the modulus number during virgin loading and unloading/reloading. A correlation is proposed between the small-strain shear modulus, $G_0$, and the modulus number, $m$. The modulus number can also be derived from static cone penetration tests, provided that the cone resistance is adjusted with respect to the mean effective stress. In a companion paper, the concepts presented in this paper are applied to data from an experimental site, where different types of seismic tests and cone penetration tests were performed.

**Keywords:** cone penetration test; constrained modulus; Poisson's ratio; sand; seismic testing; shear modulus; stiffness





## 1. Introduction

The constrained modulus is an essential parameter for assessing total and differential settlement. However, it is difficult to obtain in granular soils undisturbed samples that can be tested in the laboratory. Therefore, different types of in situ tests must be used from which the constrained modulus can be empirically derived. Various methods have been proposed in the literature concerning how the constrained modulus can be estimated from cone penetration, pressuremeter, and flat dilatometer tests. In situ tests are also needed to evaluate other important parameters affecting settlement, such as pre-consolidation stress, stress state, strength, and stiffness, as described by [1,2]. Although seismic methods can measure soil stiffness in the field and in the laboratory, their practical application has so far been limited due to difficulties correlating small-strain geotechnical parameters to those at larger strain levels (working loads).

The two main methods for assessing the constrained modulus are seismic tests and cone penetration tests. A novel concept is presented that correlates the small-strain shear modulus, $G_0$, to the tangent constrained modulus, $M_t$. Through differentiation of the relationship between the secant shear modulus and shear strain, the equivalent tangent shear modulus can be obtained. An alternative concept for estimating the constrained modulus from cone penetration tests (CPTs) is described, which uses a stress-adjusted cone resistance [3]. The application of this method for estimating the constrained modulus in silty sand was described by [4].

This paper discusses the possibilities and limitations of different types of seismic tests. The important differences between the small-strain and large-strain deformation properties of granular soils are addressed. Based on a comprehensive literature survey of seismic laboratory test results, the most important parameters affecting the shear modulus are described. Relationships are presented from which the constrained tangent modulus can be estimated based on small-strain seismic tests. The significance of these findings is discussed. The practical application of the proposed concepts is presented in a companion paper.

## 2. Design Considerations

When subjected to static loading, the design requirements for foundations on granular soils are generally governed by allowable total and differential settlements. In Europe, geotechnical design follows Eurocode 7 [5], which establishes the principles and requirements for the safety and serviceability of structures. However, regarding the prediction of settlements of spread foundations, the following caution is given in [5], 6.6.1(6): "Calculations of settlements should not be regarded as accurate. They merely provide an approximate indication." Unfortunately, only limited guidance for estimating soil compressibility can be found in the geotechnical literature for settlement analyses of foundations on silt, sand, or gravel. Hence, there is a need for more reliable methods for the determination of soil stiffness and compressibility parameters.

Based on the review of several case histories, ref. [6] pointed out that the ground exhibits higher stiffness at small strains. Dynamic measurements of shear modulus have tended to give results much higher than static values determined in the laboratory; therefore, dynamic values have frequently been discounted. However, it has been shown that the accurately determined static small-strain stiffness values are close to those measured using seismic methods. Ref. [6] suggested that correlating static to seismic deformation properties can lead to a broader application of geophysical methods for determining stiffness properties. Ref. [7] pointed out that soil stress–strain behavior is highly non-linear, which influences the selection of design parameters for simple routine geotechnical calculations.

Non-linear behavior can be characterized by soil rigidity and degree of nonlinearity [7]. These parameters can be determined from measurements of very small strain stiffness, peak strength, and failure strain. Ref. [8] demonstrated that small-strain stiffness parameters can be determined reliably from different seismic tests. He presented a constitutive framework within which geotechnical analyses can be carried out based on seismic measurements. Numerical methods can simulate highly complex loading conditions. However, an essential aspect of the numerical modeling process is the selection of a realistic constitutive model, as the chosen input parameters have a significant influence on the outcome of the analysis. It is not uncommon in design that complex analytical methods are chosen, yet over-simplified input parameters of soil stiffness are assumed. Therefore, the results of numerical analyses should be verified by parameter studies of critical input parameters (soil stiffness, stress state, and stress history) [9].

Ref. [10] showed how the deformation parameters derived from seismic tests can be used to compute the load–settlement relationship for shallow foundations on sand and clay, as well as for a single pile in residual soil. Ref. [11] outlined an approach for utilizing the results of in situ shear wave speed measurements to estimate the load–settlement behavior of shallow and deep foundations. Correlations were developed to estimate the initial foundation stiffness and, using an empirical relationship between modulus reduction and load or stress level, the settlement at various loads. Ref. [11] emphasized that the described approach should be primarily used for preliminary estimates of load–settlement behavior, to be checked against the results of more advanced design methods.

Deformation properties are usually determined in the laboratory through compression tests on undisturbed or reconstituted soil samples. However, obtaining undisturbed samples in granular soils is not feasible in conventional construction projects. Therefore, deformation properties are frequently chosen based on empirical concepts, usually derived from the results of in situ tests. Different types of field tests are available for estimating

deformation properties in granular soils. The pressure meter (PMT) and the flat dilatometer (DMT) are two methods that can measure the soil response to a lateral expansion of a cylinder (PMT) or a flat pressure cell (DMT). Empirical concepts have been developed to estimate soil modulus values from cone penetration tests (CPTs), the DMT [12] or dynamic penetration tests (SPTs). The screw plate test (SPLT) can also measure the constrained modulus, but it is not widely used. The application and evaluation of different types of in situ tests in sandy gravel have been described in detail by [13].

## 3. Seismic Tests

Soil stiffness can be measured through different types of seismic tests in the field or laboratory. For a detailed description of the application of seismic testing methods, reference is made to [14]. The shear wave and/or compression wave speed can be derived from seismic tests. Shear wave speed measurements can be made with high accuracy and are not affected by groundwater conditions. The small-strain shear modulus, $G_0$, can be derived from the shear wave speed. Alternatively, seismic laboratory tests on reconstituted soil samples can determine the shear wave speed. The main advantage of seismic laboratory tests is that the effect of strain on soil stiffness can be measured, which is not possible in the case of seismic field tests. The small-strain shear modulus, $G_0$, can be determined from

$$G_0 = C_{0s}{}^2 \rho \qquad (1)$$

where $C_{0S}$ = shear wave speed at small strain and $\rho$ = bulk density. As $G_0$ is a derived parameter, it is sensitive to the accuracy of the measured shear wave speed.

### 3.1. Seismic Field Tests

The seismic down-hole test (SCPT) is increasingly used and has the advantage of being non-intrusive, thus better reflecting in situ stress conditions. Guidelines for test execution and data interpretation have been published by ISSMGE TC 10 [15] and [16]. However, the tests must be performed and interpreted with care by experienced personnel. SCPT results can be affected by several factors, such as geotechnical conditions (soil stratification), the type of equipment, the execution of testing, and data evaluation. Ref. [17] found that in situ measurements of low-strain amplitude shear moduli generally agreed with laboratory values obtained using the RC test. Ref. [18] identified potential errors associated with two simple, straight-line ray path methods of downhole data analysis: the interval and the slope method. Seismic wave speeds obtained from the interval method can be significantly different from the correct hypothetical values. Ref. [19] pointed out the epistemic uncertainty associated with the developed shear wave profiles from SCPT. However, it is generally accepted that if guidelines and standards are followed, the shear wave speed, and thus the shear modulus at small strain, can be estimated and/or measured reliably. However, a significant limitation of seismic field tests is that measurements are usually performed at very low shear strain (typically $< 10^{-3}\%$) and low strain rates [20].

### 3.2. Seismic Laboratory Tests

A widely used method is the RC test, from which empirical relationships for estimating the small-strain shear modulus have been developed [21]. For sands, ref. [22] has proposed the following semi-empirical relationship for the estimation of the shear modulus at a small strain, $G_0$:

$$G_0 = \frac{625}{0.3 + 0.7e^2} OCR^k \left( \sigma_0' \sigma_r \right)^{0.5} \qquad (2)$$

where $e$ = void ratio; $OCR$ = overconsolidation ratio; $k$ = empirical constant, which depends on the plasticity index (*PI*); $\sigma_0'$ = the mean effective stress; and $\sigma_r$ is reference stress (100 kPa). The $G_0$ in sands is a function of the square root of the mean effective stress and, thus, also of the vertical effective stress. The mean effective stress $\sigma_0'$ can be determined from

$$\sigma_0' = \frac{(1 + 2\,K_0)}{3}\,\sigma_v' \tag{3}$$

where $K_0$ = coefficient of lateral earth stress at rest (effective stress) and $\sigma_v'$ = vertical effective stress. [22] has suggested the following relationship for estimating the parameter $k$ from *PI*:

$$k = 0.006PI + 0.045 \tag{4}$$

where *PI* = plasticity index. According to Equations (2) and (4), in granular soils, $G_0$ is not strongly affected by *OCR* if *PI* is <10%. The general validity of the relationship given by Equation (2) for granular soils has been confirmed by others [17,23–25].

### 3.3. Effect of Strain Rate

The fact that the shear modulus determined from seismic tests is generally higher than that from static tests has frequently been attributed to "dynamic effects". However, seismic field or laboratory tests are usually performed under very low strain (<10$^{-3}$%), at which the loading rate is slow and comparable to that during conventional static tests [20,26–28]. During a seismic test, the soil particles are vibrated at the granular structure [29]. If the strain level is smaller than the "elastic threshold strain", elastic deformations occur at grain contacts, and the fabric remains unchanged. Ref. [30] found that in granular soils (sands), the strain rate effect on the soil stiffness measured under small strain is negligible. Under small strains, even if fully saturated, no increase in pore water pressure is measured as the loading rate is so slow that drainage can occur through the voids of granular soils. As the strain rate during a seismic test is very low, at least at a low-shear-strain level, it can be misleading to describe a seismic test as a "dynamic test". This term should be reserved for methods where the loading rate and inertia effects are significant.

### 3.4. Effect of Strain Level

An essential aspect of seismic analyses is the effect of shear strain on soil stiffness. Soil stiffness decreases with increasing strain, particularly in granular soils (silt, sand, or gravel). The following RC test on a silty sand illustrates stiffness degradation. The soil sample consisted of silty sand with a plasticity index, *PI* = 14. The bulk density at a water content of 32% was $\rho = 1.870\ \text{kg/m}^3$. Figure 1 shows the variation in shear wave speed, $C_S$, with shear strain, $\gamma$. The strain-dependent secant shear modulus, $G_S$, can be derived from the shear wave speed, $C_s$, according to the following relationship:

$$G_s = C_S{}^2\rho \tag{5}$$

where $\rho$ = bulk density of the sample. The derived secant shear modulus, $G_S$, is also shown in Figure 1.

Both the shear wave speed and the secant shear modulus are strain-dependent. At 0.0002% shear strain, the shear wave speed is 206 m/s and the maximum shear modulus is 76 MPa. When exceeding about 0.01% shear strain, the shear wave speed and the shear modulus start to decrease markedly. At 0.1%, a strain level at which conventional laboratory measurements obtain the first data point, the secant shear modulus (24 MPa) is only 32% of its maximum value (76 MPa).

The decrease in secant shear modulus, $G_S$, as a function of shear strain, $\gamma$, can be expressed by the following relationship [20]:

$$\frac{G_S}{G_0} = \frac{1}{1 + \alpha\gamma\left(1 + 10^{-\beta\gamma}\right)} \tag{6}$$

where $\alpha$ and $\beta$ are empirically determined parameters. For fine-grained soils, and based on an examination of modulus reduction curves for different soils, the following relationship for estimating the parameters $\alpha$ and $\beta$ was proposed:

$$\alpha = \frac{1 - m_2 PI^{n_2}}{m_2 PI^3 \gamma_2 \left(1 + 10^{-\beta \gamma_2}\right)} \tag{7}$$

$$\beta = \frac{\log\left(\frac{m_2}{m_1}\right) + (n_2 - n_1)\log(PI) + \log\left(\frac{\gamma_2}{\gamma_1}\right)}{(\gamma_2 - \gamma_1)} \tag{8}$$

where $\gamma_1$ and $\gamma_2$ are two strain levels chosen to define modulus degradation and $n_1$, $n_2$, $m_1$, and $m_2$ are empirically determined parameters, respectively. Exponents $\alpha$ and $\beta$ are shown in Figure 2 as a function of the plasticity index, *PI*, from which typical values can readily be chosen.

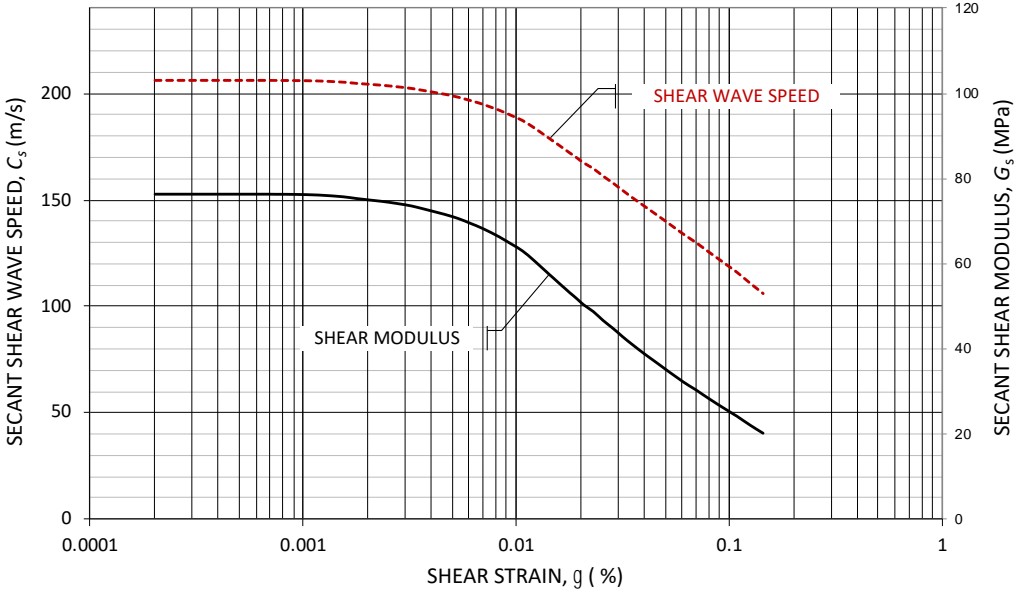

**Figure 1.** Decrease in shear wave speed (left axis) and shear modulus (right axis) with shear strain, determined from a resonant column test on silty sand.

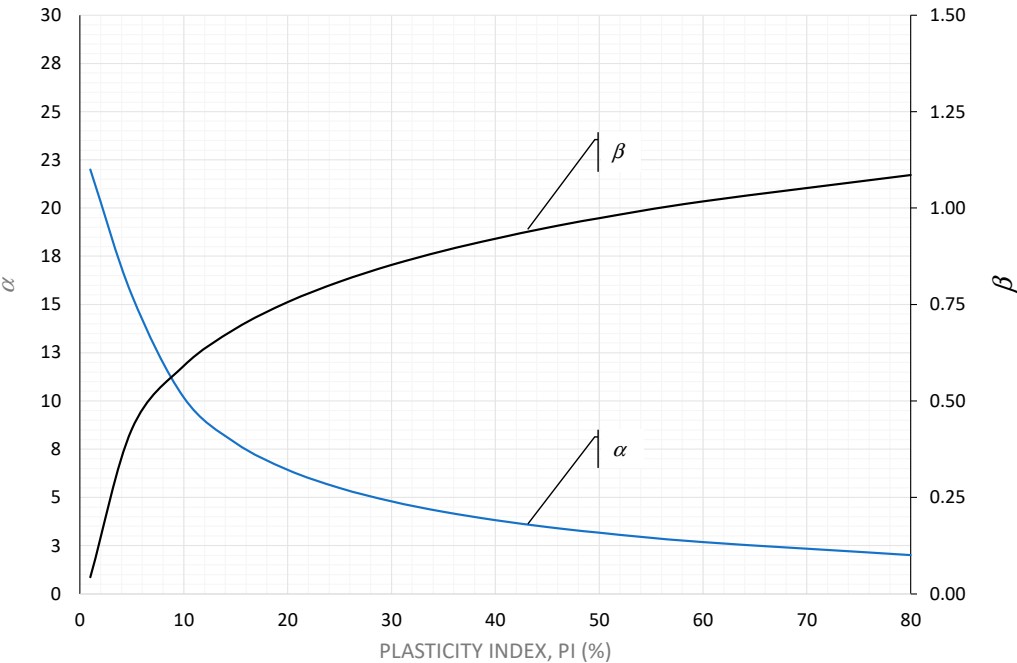

**Figure 2.** Empirical parameters as a function of plasticity index, *PI*, according to Equations (7) and (8).

Based on an evaluation of RC test data for a wide range of soils, the values of $m_1$ and $m_2$ presented in Table 1 gave the best fit of the modulus degradation curve at strain levels $\gamma_1$ = 0.1% and $\gamma_2$ = 0.5%, respectively [20].

**Table 1.** Empirical parameters, $m_1$ and $m_2$, for estimation of exponents $\alpha$ and $\beta$ at 0.1 and 0.5% shear strain, respectively.

| | $\gamma_1$: 0.1% | | $\gamma_2$: 0.5% |
|---|---|---|---|
| $m_1$: | 0.1273 | $m_2$: | 0.0265 |
| $n_1$: | 0.4198 | $n_2$: | 0.6388 |

Assuming the values given in Table 1, the parameters $\alpha$ and $\beta$, obtained for granular soils with *PI* ranging from 0 to 10%, are presented in Table 2. Substituting $\alpha$ and $\beta$ given in Table 2 into Equation (6), the secant shear modulus degradation curves for soils with *PI* ranging from 0 to 10% shown in Figure 3 are obtained.

**Table 2.** Recommended parameters $\alpha$ and $\beta$ for granular soils according to Table 1, cf. Figure 2.

| *PI* (%) | 0 | 5 | 10 |
|---|---|---|---|
| | Low | Medium | High |
| $\alpha$ | 22.00 | 15.50 | 10.18 |
| $\beta$ | 0.04 | 0.43 | 0.59 |

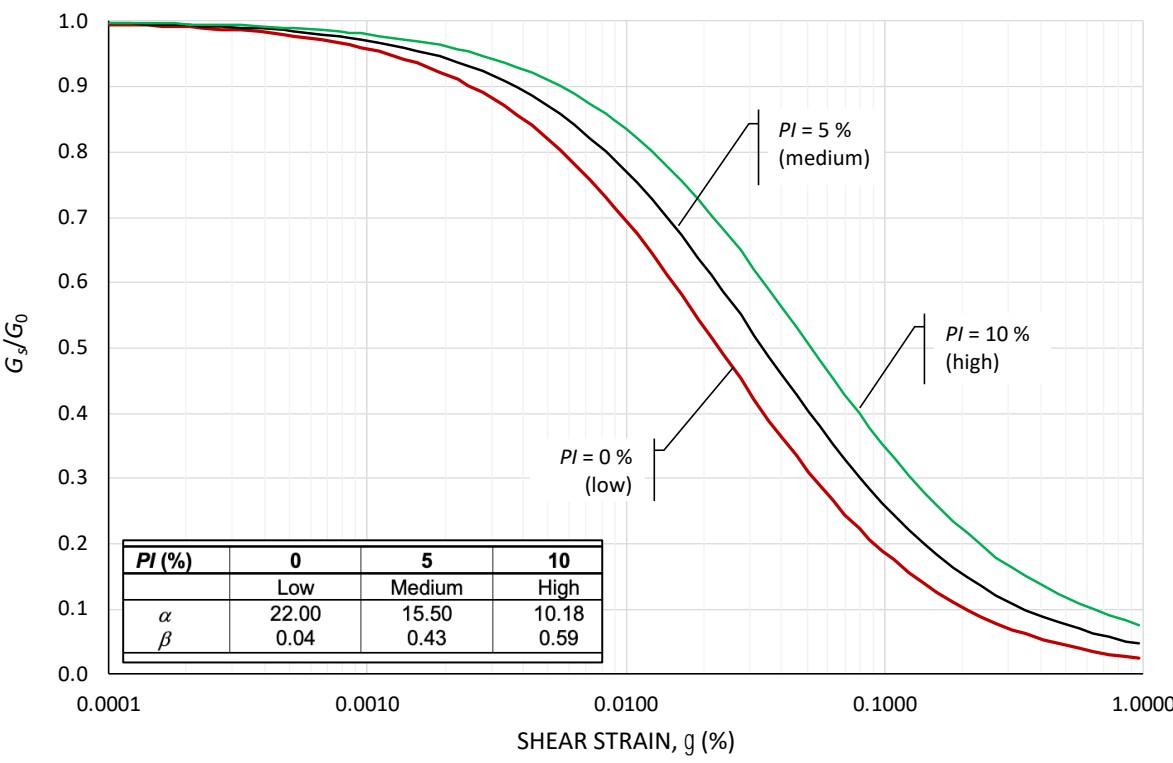

**Figure 3.** Secant shear modulus degradation curve for sands and silt (0 < *PI* < 10) according to [20], cf. Equation (6) and Table 2.

The secant modulus degradation curves agree with other studies [24,25,31]. In subsequent sections of this paper, the three degradation curves of the secant modulus will be referred to as "low" (*PI* = 0%), "medium" (*PI* = 5%), and "high" (*PI* = 10%), respectively. For important design projects, the parameters recommended in Table 2 must be verified by laboratory tests.

## 4. Review of Shear Modulus Degradation Data

Ref. [32] studied the effect of different geotechnical parameters (void ratio, degree of saturation, and plasticity index) on the secant shear modulus reduction for sand and silt. The degree of saturation, $S_r$, had little influence on modulus reduction in the range between 0.1 and 0.5% shear strain. However, the modulus reduction coefficient, $G_s/G_0$, increased slightly (about 5%) with an increasing void ratio (from $e = 0.3$ to 0.7).

As part of this study, the secant shear modulus degradation data of granular soils obtained by the authors listed in Table 3 have been reviewed. Most of the tests performed were RC or TS tests. The tests investigated the effect of different factors, such as mean effective stress, relative density, plasticity index, and grain size, on soil stiffness degradation.

**Table 3.** Investigations showing the influence of different parameters and soil types on shear modulus degradation.

| Investigated Parameters/Soil Types | Value | Reference |
|---|---|---|
| Mean effective stress (clean and silty sands) | 25 kPa<br>100 kPa<br>400 kPa | Adopted from [33] |
| Average of different sands at constant mean effective stress | 100 kPa | [34] |
| Relative density | 30–90% | [24] |
| Mean effective stress (sand) | 50 kPa<br>150 kPa<br>400 kPa<br>800 kPa<br>2000 kPa<br>4000 kPa | [35] |
| Silty, clayey sand | $PI = 14\%$ | [27] |
| Non-plastic soils; one loading cycle, $N$ | $PI = 0\%$ | [31] |
| Gravelly soils | Lower boundary<br>Best fit<br>Upper boundary | [36] |
| Plasticity index | $PI = 1\%$<br>$PI = 5\%$<br>$PI = 10\%$ | [20] |

Secant shear modulus degradation $G_s/G_0$ data for sandy soils, from the investigations listed in Table 3, are presented in Figure 4 and compared with the correlations according to Equation (6), as shown in Figure 3. The secant shear modulus increases with mean effective stress. Otherwise, the degradation curves of all tests fall within a relatively narrow range and are inside the boundaries shown in Figure 3. The variation in $G_s/G_0$ data decreases with increasing shear strain and falls within a narrow range at shear strains >0.5%.

### 4.1. Confining Stress

Ref. [35] compiled a database showing the secant shear modulus degradation curves of 454 tests as a function of effective confining stress. In addition, the authors performed their own RC tests on sands. They formulated an S-shaped curve of secant shear modulus degradation for different values of the effective confining stress, fitting a modified hyperbolic relationship. The mean effective confining stress varied between 50 kPa and 4000 kPa. For normal geotechnical applications, the confining stress typically ranges between 50 and 150 kPa. However, much higher confining stresses can occur in earth dam foundations. The results of their compilation of data are shown in Figure 5, together with the reference curves from Figure 3. At a low confining stress, secant shear modulus degradation starts at a lower strain level compared to higher confining stress. The modulus degradation

curves are, at confining stresses ranging from 50 to 400 kPa, in good agreement with the boundaries (low, medium, and high), shown in Figure 3. Ref. [35] also found that sands with more dispersed particle sizes begin to lose their linear elastic stiffness at a lower strain than is the case for more uniform sands.

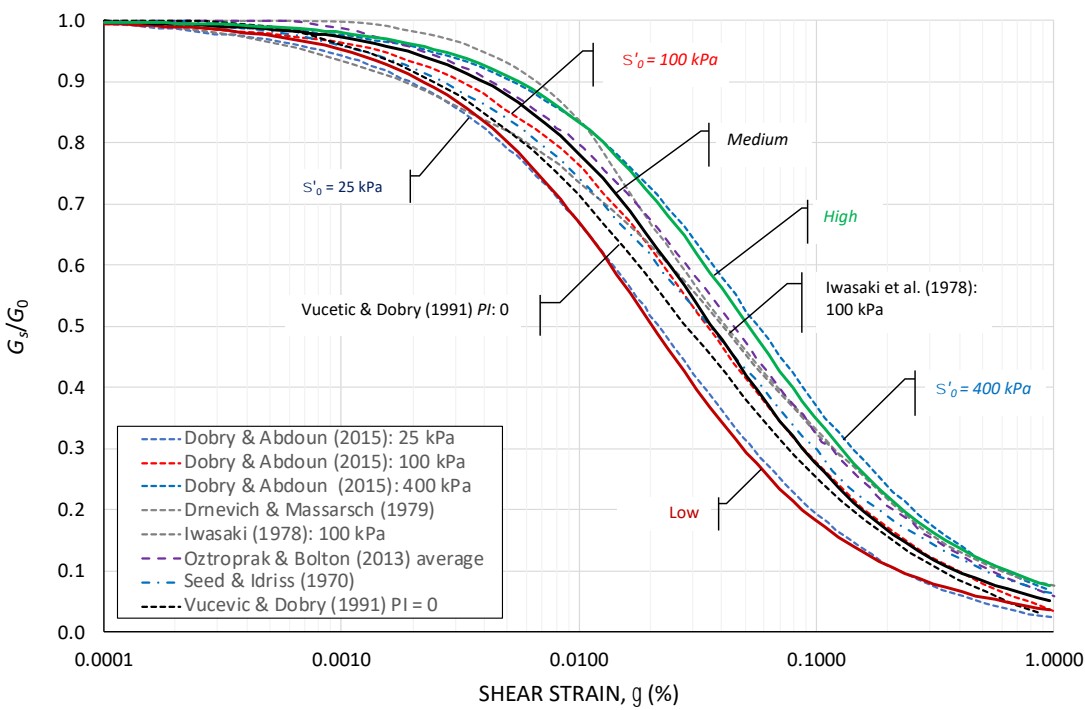

**Figure 4.** Compilation of secant shear modulus degradation curves from investigations listed in Table 3, and compared with reference curves (low, medium, and high) shown in Figure 3 [26,31,34,35,37,38].

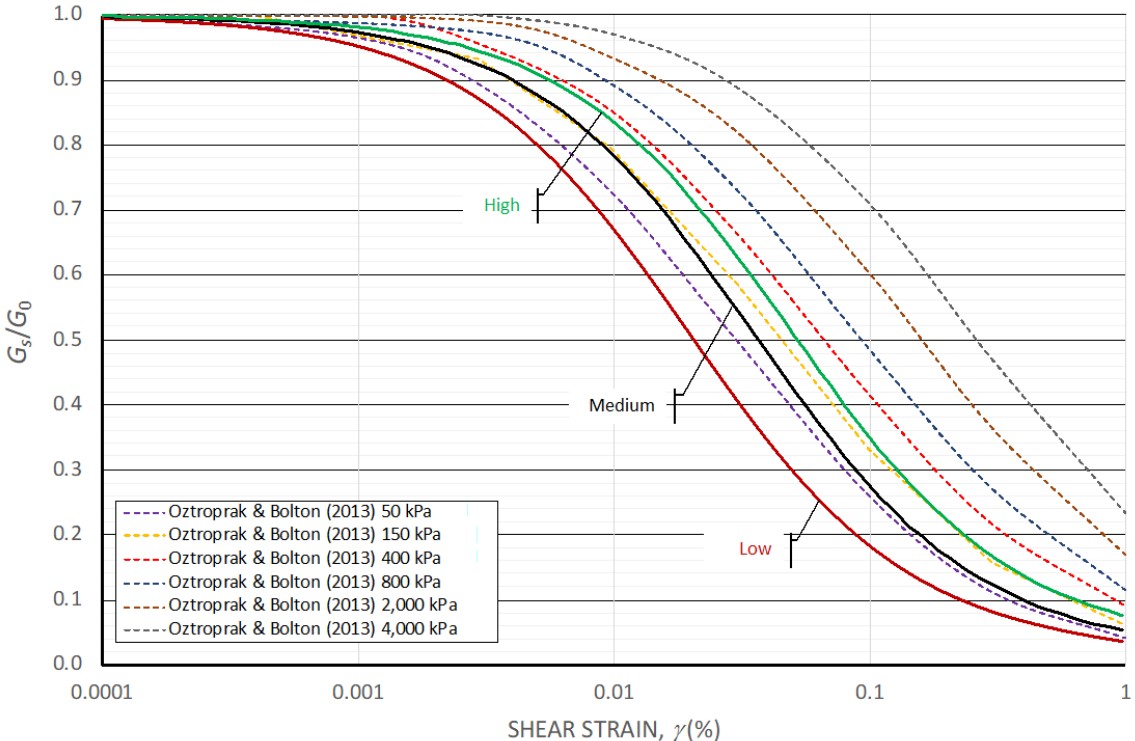

**Figure 5.** Effect of confining stress on secant shear modulus degradation of sand [35].

The results from a wide variety of investigations suggest that the secant shear modulus degradation relationships of sandy soils fall within a relatively narrow range, especially at larger shear strain (>0.5%). There is also a clear trend that an increase in confining stress moves the shear strain degradation curve towards the upper range of curves shown in Figure 3.

### 4.2. Particle Size

Other important parameters affecting secant shear modulus degradation are particle size and grain size distribution [22,24,39]. Ref. [36] compiled the results of 15 investigations on modulus degradation for gravelly soils and combined these with their own test results. Figure 6 shows the findings by [36] for gravelly soils, which they divided into three groups: lower-boundary, average, and upper-boundary values. For comparison, the range for sandy soils, as presented in Figure 3, is also shown.

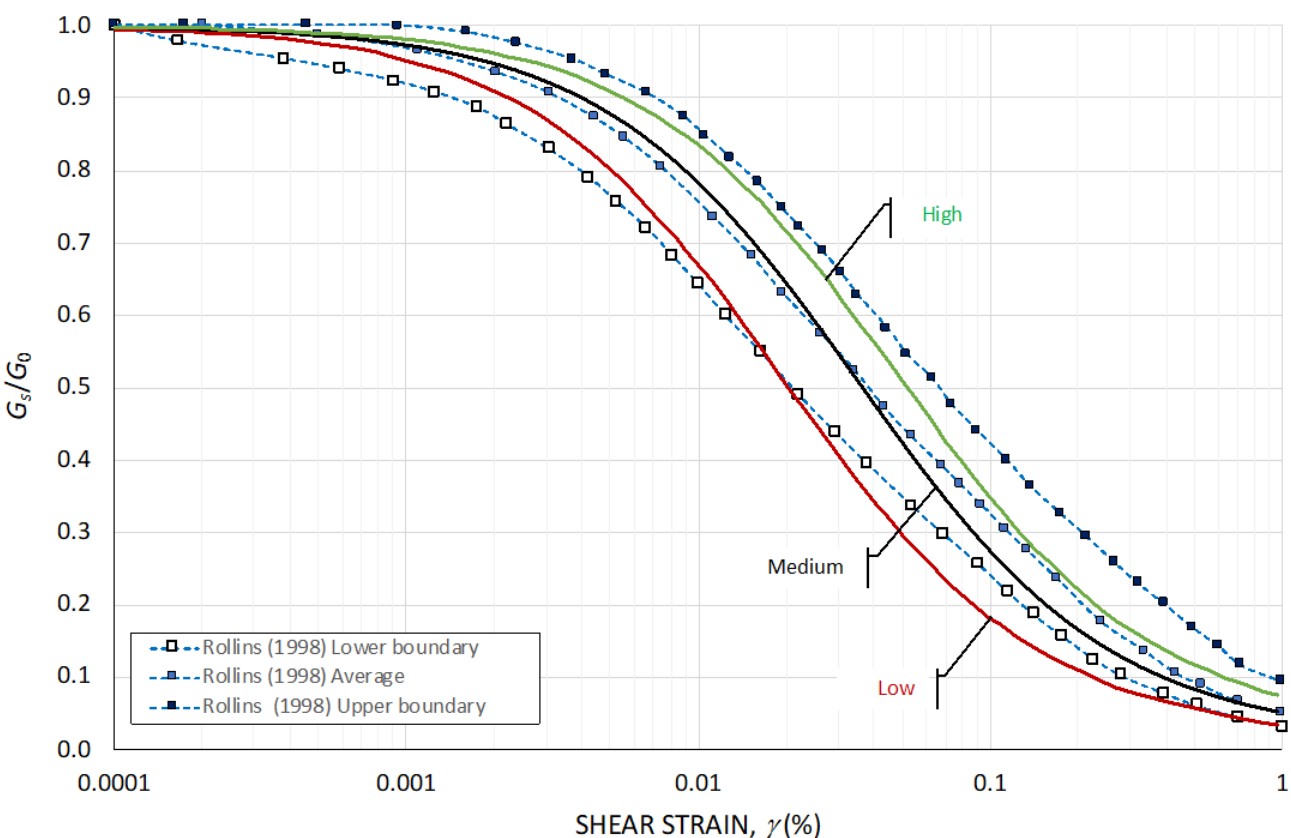

**Figure 6.** Modulus degradation of gravelly soils based on [36], together with boundary ranges stated in Figure 3.

Ref. [36] concluded that the results from 15 investigations of gravelly soils were almost independent of fines content, gravel content, and relative density. As the confining stress increases, the curves move closer to the upper range of the data, as confirmed by the results presented in Figure 5. It can be concluded that gravelly soils, on average, behave similarly to sandy soils. Only the upper boundary for gravel deviates from the modulus degradation curves of sands.

### 4.3. Degradation of Secant Shear Modulus

In the case of sandy soils, degradation of the secant shear modulus as a function of shear strain within a strain range from 0.0001 to 1% can be mathematically approximated by Equation (6). The parameters, $\alpha$, and $\beta$, can be determined for different soils from Figure 2 or Tables 1 and 2. However, it may be sufficient for many practical applications—at least at

a preliminary design stage—to use the empirical values proposed in Table 4. The exponents $\alpha$ and $\beta$, which define the shape of the secant shear modulus degradation curve, were determined based on data shown in Figure 4. As a first estimate for granular soils, it is recommended to use the following values for sand (medium) in Table 4.

**Table 4.** Typical values of exponents $\alpha$ and $\beta$ for different soil types, cf. Equation (6).

| Soil Type | $\alpha$ | $\beta$ | Reference |
|---|---|---|---|
| Sand low | 25 | 1 | Low (this investigation) |
| Sand medium | 14 | 0.5 | Medium (this investigation) |
| Sand high | 10 | 0.6 | High (this investigation) |
| Sand ($PI = 0$) | 20 | 4.5 | [31] |
| Gravel loose | 45 | 40 | [36] low |
| Gravel average | 20 | 12 | [36] average |
| Gravel dense | 8.5 | 2 | [36] high |
| $PI = 1$ | 22 | 0.04 | |
| $PI = 5$ | 15 | 0.4 | |
| $PI = 10$ | 10 | 0.6 | [20] |
| $PI = 15$ | 8 | 0.7 | |
| $PI = 20$ | 6 | 0.8 | |

## 5. Tangent Modulus Concept for Settlement Analyses

Although not widely used outside Europe, the settlement analysis method proposed independently by [40,41] is a simple yet powerful concept. It is compatible with other, more well-known settlement analysis methods and is summarized in [42]. For a detailed description of the tangent modulus method and its practical application for settlement analyses, reference is made to [43] and his "Rankine lecture" [44]. The following sections describe the tangent modulus concept for normally consolidated and swelling/reloaded soils.

The tangent constrained modulus, $M_t$, is the ratio between the change in stress and the change in strain

$$M_t = \frac{d\sigma}{d\varepsilon} = m\,\sigma_r \left( \frac{\sigma'_v}{\sigma_r} \right)^{(1-j)} \tag{9}$$

where $d\sigma$ = change in stress; $d\varepsilon$ = change in strain; $m$ = modulus number (dimensionless); $\sigma_r$ = reference stress (equal to 100 kPa); $\sigma'_v$ = vertical effective stress; and $j$ = stress exponent. Integrating Equation (9) yields the following general relationship for calculating the strain, $\varepsilon$, of a soil layer:

$$\varepsilon = \frac{1}{m\,j} \left[ \left( \frac{\sigma'_{v1}}{\sigma_r} \right)^j - \left( \frac{\sigma'_{v0}}{\sigma_r} \right)^j \right] \tag{10}$$

where $\sigma'_{v0}$ = vertical effective stress prior to loading and $\sigma'_{v1}$ = vertical effective stress after loading.

An important aspect of the tangent modulus concept (Equation (10)) is the selection of realistic input values, that is, the stress exponent, $j$, and the modulus number, $m$. The stress exponent defines the shape of the stress–compression curve. In the case of granular soils (sand), two cases of the stress exponent are of relevance, $j = 0.5$ (normally consolidated—virgin loading) and $j = 1.0$ (unloading–reloading). The modulus number, which defines soil stiffness, is more difficult to assess in the case of granular soils, as it is not possible to obtain undisturbed samples. Based on data by [43,44], Table 5 summarizes, for different soil types, typical values of $j$ and the range and average value of $m$, respectively.

**Table 5.** Typical stress exponents and modulus numbers for granular soils based on [43,44].

| Soil Type | Stress Exponent, *j* | Modulus Number, *m* | |
| | | Range | Average |
|---|---|---|---|
| Till, very dense to dense | 1.0 | 1000–300 | 650 |
| Gravel | 1.0 | 400–40 | 220 |
| Sand | | | |
| dense | 1.0 | 400–250 | 325 |
| compact | 0.5 | 250–150 | 200 |
| loose | 0.5 | 150–100 | 125 |
| Silt | | | |
| dense | 1.0 | 200–80 | 140 |
| compact | 0.5 | 80–60 | 70 |
| loose | 0.5 | 60–40 | 50 |

*5.1. Modulus Number—Virgin Loading*

Ref. [44] updated the values of his previously proposed modulus number, *m*, for normally consolidated silt and sand as a function of porosity, *n*, which in this paper has been converted to the more widely used void ratio, *e*. Figure 7 shows the range of modulus numbers for normally consolidated silts and sands as a function of void ratio for different soil densities.

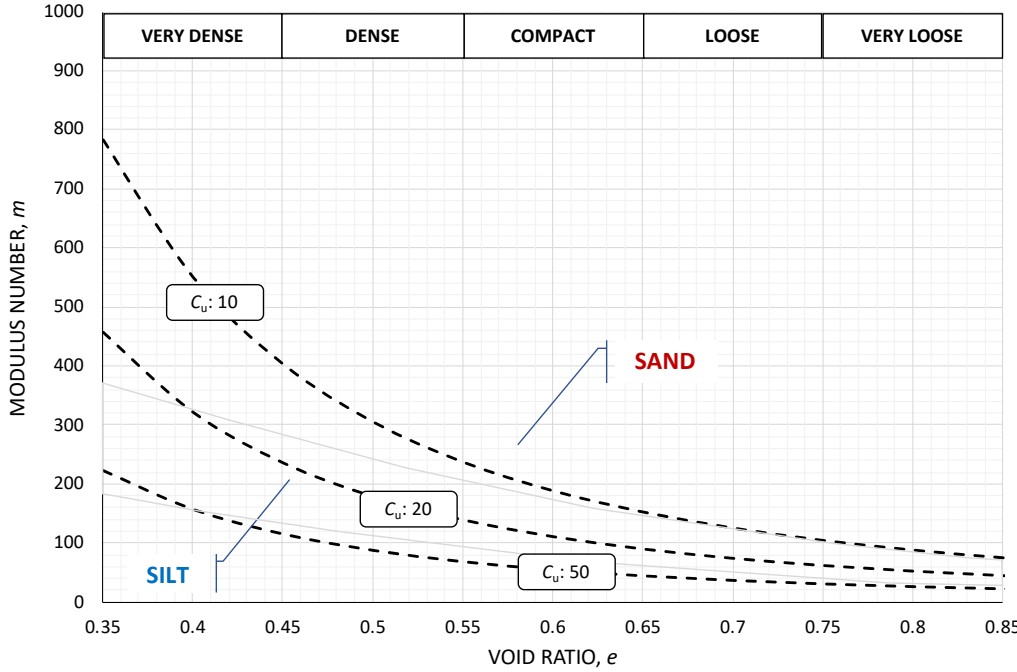

**Figure 7.** Typical values of modulus numbers for normally consolidated sand and silt, after [44], and $C_U$ ($d_{50} < 5$ mm) according to Equation (11).

Ref. [45] performed a comprehensive laboratory study on the compressibility of granular soils, re-analyzing compression tests presented in the literature. In addition, he performed large-diameter oedometer and ring compressometer tests on granular soils, ranging from fine sand to coarse gravel and crushed stone. He found that the modulus number, *m*, is strongly dependent on the initial void ratio, $e_0$, and grain size, defined by the coefficient of uniformity, $C_U$ ($d_{60}/d_{10}$):

(a) Oedometer tests on sand ($d_{50} < 5$ mm)

$$m = 295C_U{}^{-0.78}e_0{}^{-2.64} \qquad (11)$$

(b)    Ring compressometer tests on coarse-grained soils ($d_{50} > 10$ mm)

$$m = 271C_U^{-0.71}e_0^{-3.72} \tag{12}$$

These results expand the database for estimating the modulus number to cover a wider range of granular soils, showing that in addition to the void ratio, *e*, the uniformity coefficient, $C_U$, is also of significance when estimating the modulus number, *m*. It can be concluded that the two independent investigations by [44,45] arrive at similar values of the modulus number for normally consolidated sands and silts, cf. Figure 7.

Methods have been proposed in the literature where the modulus number is correlated directly to the cone resistance [46] without considering the effect of depth (confining stress) as discussed above.

*5.2. Modulus Number—Unloading and Re-Loading*

Stress history (preloading) affects the tangent constrained modulus. During unloading and subsequent reloading, only a minor portion of the initial strain occurs. As a result of preloading, both the stress exponent (shape of the reloading curve) and the modulus number (soil stiffness) increase. For practical purposes, it can be assumed that the average tangent constrained modulus during unloading (swelling) is approximately the same as the average modulus during reloading. Unfortunately, only limited information is available in the literature that quantifies the effect of preloading on the tangent modulus. Granular soils behave essentially elastically during reloading, with the stress exponent, *j*, approaching 1.0 [40,41]. Ref. [45] measured the tangent constrained modulus during virgin loading and unloading (swelling) and found that the unloading modulus is almost independent of soil density for all tested sandy soils. The tangent modulus during unloading, $M_{tu}$, as a function of vertical effective stress, is shown in Figure 8. It depends strongly on the vertical effective stress for all tests.

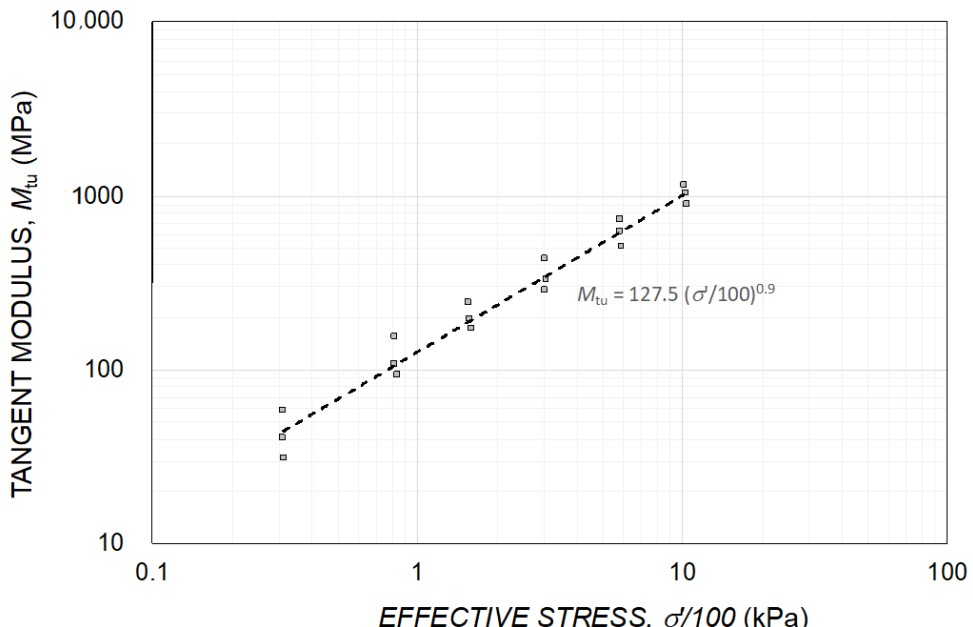

**Figure 8.** Tangent constrained modulus during unloading as a function of vertical effective stress, data re-analyzed from [45].

The unloading test data reported by [41] were analyzed by [47]. Figure 9 shows the relationship between the modulus number ratio, $m_u/m$, and the modulus number, *m*, at virgin loading (normally consolidated), where $m_u$ is the modulus number determined from the unloading test. The increase in the unloading modulus number is most significant at

low values of the modulus number (loose sands). The correlation between the $m_u/m$ ratio and $m$ at virgin loading shown in Figure 9 can be expressed by the following relationship:

$$\frac{m_u}{m} = 225m^{-0.76} \tag{13}$$

Equation (13) can be used to estimate the unloading modulus for different values of the virgin modulus number. For example, in loose sand ($m$~100), the unloading modulus ratio $m_u/m \approx 7$. In the case of an initially denser sand ($m$~300), the unloading modulus ratio is lower, $m_u/m \approx 3$. When the virgin modulus increases towards $m$~1250, the $m_u/m$-ratio approaches 1. Thus, the increase in the $m_u/m$ ratio is more pronounced in loose granular soils than in dense soils. Refs. [41,44] have proposed that the constrained modulus at unloading and reloading can be estimated to be about 5 to 8 times higher than the virgin loading modulus. The information in Figure 9 is thus in good agreement with experience.

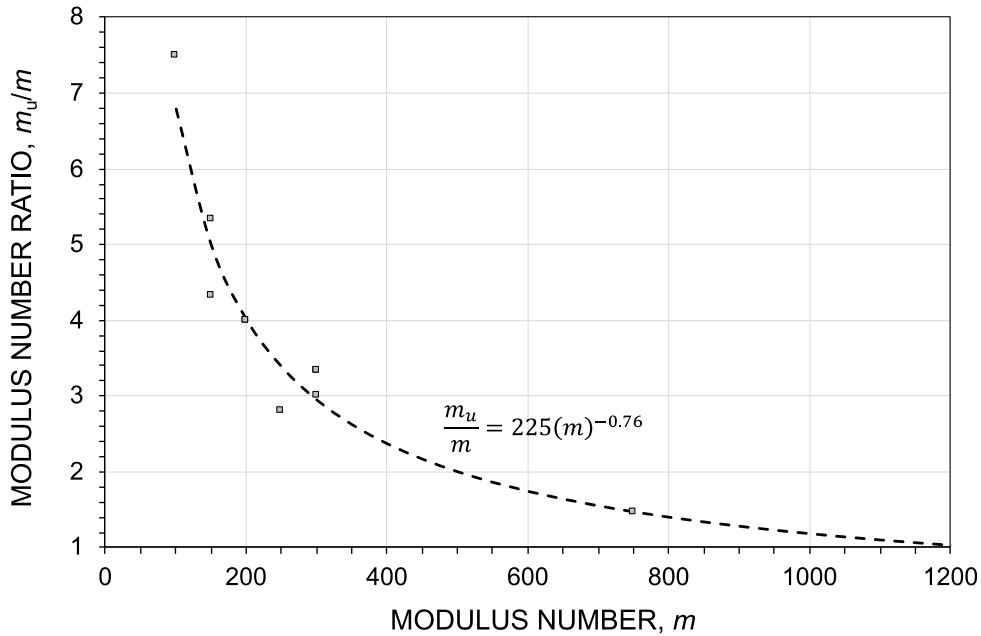

**Figure 9.** Unloading modulus number ratio as a function of modulus number for virgin loading; data from [41].

*5.3. Modulus Number from CPT*

There is a need for estimating $m$ from in situ tests such as the CPT or CPTU. Based on extensive experience from soil compaction projects, a method was proposed by [3] and refined by [47], where the modulus number can be estimated from a stress-adjusted cone resistance, $q_{CM}$

$$m = a\left(\frac{q_{CM}}{\sigma_r}\right)^{0.5} \tag{14}$$

where $a$ = empirical modulus factor and $\sigma_r$ = reference stress (100 kPa). The stress-adjusted cone resistance can be determined from the following relationship:

$$q_{CM} = q_c\left(\frac{\sigma_r}{\sigma'_0}\right)^{0.5} \tag{15}$$

where $q_c$ = cone resistance; $\sigma'_0$ = mean effective stress. The modulus factor, $a$, reflects soil type as shown in Table 6.

**Table 6.** Modulus factor, *a*, for granular soils [47].

| Soil Type | | Empirical Modulus Factor, *a* |
|---|---|---|
| Silt | | |
| | organic soft | 7 |
| | loose | 12 |
| | compact | 15 |
| | dense | 20 |
| Sand | | |
| | silty loose | 20 |
| | loose | 22 |
| | compact | 28 |
| | dense | 35 |
| Gravel | | |
| | loose | 35 |
| | compact | 40 |
| | dense | 45 |

## 6. Relationship between Shear Modulus and Constrained Modulus

Estimating the soil stiffness in granular soils is a challenging task, as this must be based on in situ tests. Seismic tests have many advantages when investigating geotechnical properties. However, in the past, there have been difficulties in correlating the small-strain shear modulus, $G_0$, with other modulus values.

This section of the paper outlines a concept of how the constrained modulus at large strain can be estimated from the tangent modulus at small strain obtained from seismic tests. As a first step, a relationship defining the degradation of the secant modulus with strain is presented. The relationship, valid for the secant shear modulus, is derived to obtain the relationship between the tangent shear modulus with shear strain as a function of plasticity index. Poisson's ratio is needed to correlate the shear modulus with the constrained modulus. However, as is shown, Poisson's ratio is not a constant value, as frequently assumed, but varies with strain. As a final step, the derived constrained modulus can be correlated with the tangent modulus obtained from conventional laboratory tests.

Efforts have been made to correlate $G_0$ to the tangent constrained modulus, $M_t$ [11,48]. At very low strain, the secant and tangent moduli are identical but start to deviate with increasing strain. Ref. [49] have shown that the secant modulus can be converted to an equivalent tangent modulus by differentiation of the stress–strain relationship. This aspect is especially important in granular soils when comparing the secant shear modulus at large strain with an equivalent tangent constrained modulus.

### 6.1. Tangent Shear Modulus

The degradation of soil stiffness is usually expressed in terms of the secant shear modulus, $G_S$. However, in the case of incremental loading, the tangent shear modulus, $G_t$, should be used, which can be determined by differentiation of Equation (6). For this purpose, a modulus degradation factor, $R_G$, is introduced:

$$R_G = \frac{G_s}{G_0} \tag{16}$$

The following relationship exists between shear stress and shear strain:

$$\tau = G_0 \gamma R_G(\gamma) \tag{17}$$

Now, the shear stress, $\tau$, can be differentiated with respect to shear strain, $\gamma$:

$$\frac{\partial(\tau)}{\partial \gamma} = \frac{\partial(G_0 \gamma R_G(\gamma))}{\partial \gamma} \tag{18}$$

Moreover, the modulus degradation relationship defined by Equation (18) can be substituted into Equation (6)

$$R_G = \frac{G_S}{G_0} = \frac{1}{1 + \alpha\gamma(1 + 10^{-\beta\gamma})} \tag{19}$$

and differentiation with respect to $\gamma$

$$\frac{\partial(\tau)}{\partial\gamma} = \frac{\partial}{\partial\gamma}\left(\frac{G_0\gamma}{1 + \alpha\gamma(1 + 10^{-\beta\gamma})}\right) \tag{20}$$

yields the general relationship between the tangent shear modulus, $G_t$, and the small-strain shear modulus, $G_0$:

$$\frac{G_t}{G_0} = \frac{10^{\beta\gamma}(\alpha\beta\gamma^2\log(10) + 10^{\beta\gamma})}{(\alpha\gamma(10^{\beta\gamma} + 1) + 10^{\beta\gamma})^2} \tag{21}$$

This relationship between $G_t$ and $G_0$ is valid for all soil types when appropriate values of $\alpha$ and $\beta$ are chosen. The following approximate relationship between $G_t$ and $G_0$ as a function of $\gamma$ can be obtained by substituting typical values for sand (medium) according to Table 4: $\alpha = 14$ and $\beta = 0.5$.

$$G_t = G_0\frac{(10^{0.5\gamma})(7\gamma^2 + 10^{0.5\gamma})}{[14\gamma(1 + 10^{0.5\gamma}) + 10^{0.5\gamma}]^2} \tag{22}$$

For medium sand ($\alpha = 14$, $\beta = 0.5$) at a shear strain level of $\gamma = 0.25\%$, which is used to represent static loading, the following simple relationship between $G_t$ and $G_0$ is obtained:

$$G_t = 0.0262G_0 \tag{23}$$

It is apparent that at a strain level reflecting static loading, the tangent shear modulus is only a fraction of the small-strain tangent shear modulus.

## 6.2. Poisson's Ratio

Poisson's ratio, $v$, is the negative ratio of transversal strain to the axial strain in an elastic material, which is subjected to a uniaxial stress. However, granular soils are not elastic materials and behave non-linearly even at small strains. Poisson's ratio depends on several parameters, such as the strain level, confining stress, saturation and drainage conditions, and degree of cementation. Ref. [50] defined an "elastic threshold strain" ($<10^{-4}\%$) at which it can be assumed that deformations take place at inter-particle contacts. During the early stages of a test, $v$ typically has values between 0.1 and 0.2 [51]. Ref. [52] used a modified RC test for measuring $v$ under random noise excitation. The device permitted measurements at very low shear strains ($\gamma \sim 10^{-6}\%$). A dry silica sand ($d_{50} = 0.44$ mm, $C_U = 0.96$) was tested. Poisson's ratio was computed for a specimen subjected to incremental confinement by exciting the first flexural and torsional mode at the same volumetric average strain levels. Values of Poisson's ratio were small, $v < 0.07$, and increased with pressure, as shown in Figure 10.

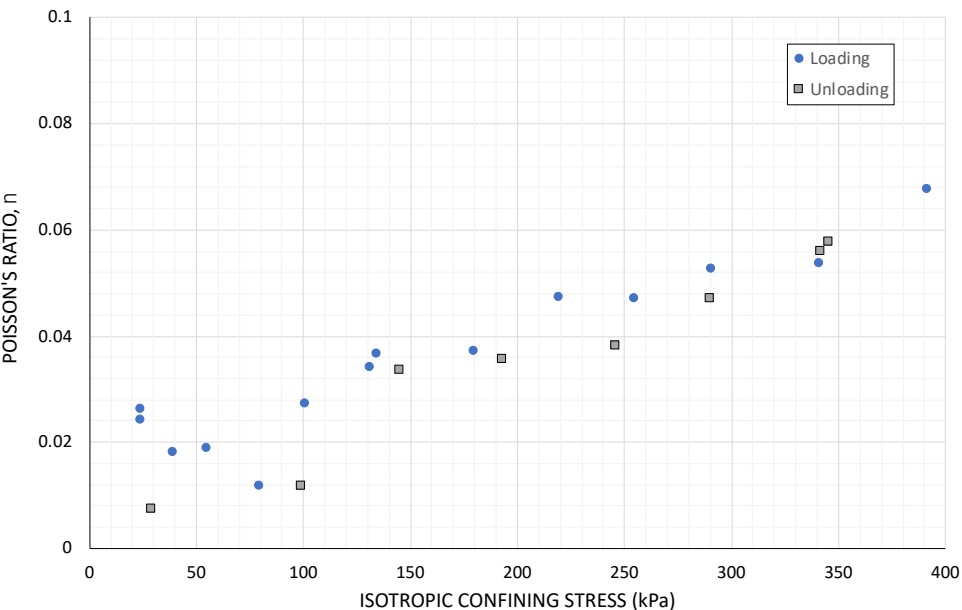

**Figure 10.** Poisson's ratio at very low shear strain (<10$^{-6}$%) computed from the flexural and torsional modes—effect of confinement; data from [52].

In natural soils, the drained Poisson ratio at low shear strain ($\gamma$ < 0.001%) is typically between 0.05 and 0.15. Ref. [53] conducted RC tests on clayey sand (*PI* = 19%). They reported values ranging between 0.1 and 0.25 for shear strains <0.001%. Cementation in sandy soils also affects $v$, indicating typical values of 0.3. The influence of grain size distribution on $v$ was investigated by [54] through RC tests with additional P-wave measurements. They found that in the case of dry quartz sands, $v$ increased with a higher coefficient of uniformity, $C_U$ ($d_{60}/d_{10}$), but did not depend significantly on mean grain size, $d_{50}$. Despite a low shear strain amplitude (<10$^{-4}$%), surprisingly high values of $v$ were obtained, ranging between 0.28 and 0.33; cf. Figure 10. Poisson's ratio for sand becomes constant for large strains (approaching failure), which in drained conditions is approximately $v$ = 0.3. In undrained conditions, the stiffness of water is much greater than the skeletal stiffness of soil particles, and the undrained Poisson's ratio tends toward 0.5. However, in the case of seismic tests, especially at very low shear strain, the loading rate is very slow, permitting at least partial drainage even in fully saturated granular soils [20,27,30].

Ref. [49] have suggested the following relationship between $v$ and the ratio between the tangent shear modulus, $G_t$, and the shear modulus at very low strain, $G_0$:

$$v = F \frac{(1 + v_0) - \frac{G_t}{G_0}(1 - 2 v_0)}{2(1 + v_0) + \frac{G_t}{G_0}(1 - 2 v_0)} \tag{24}$$

where $v_0$ = initial Poisson's ratio, which is set at 0.1. According to this relationship, Poisson's ratio varies between 0.1 and 0.5. However, based on the above-reported test results for granular soils and drained conditions, the value of $v$ should be limited to 0.33. Therefore, a scaling factor, $F$ = 0.65, has been added, which also reduces $v_0$ to 0.05, and is in better agreement with the data shown in Figure 10. The variation in Poisson's ratio as a function of shear strain according to Equation (24) is shown in Figure 11.

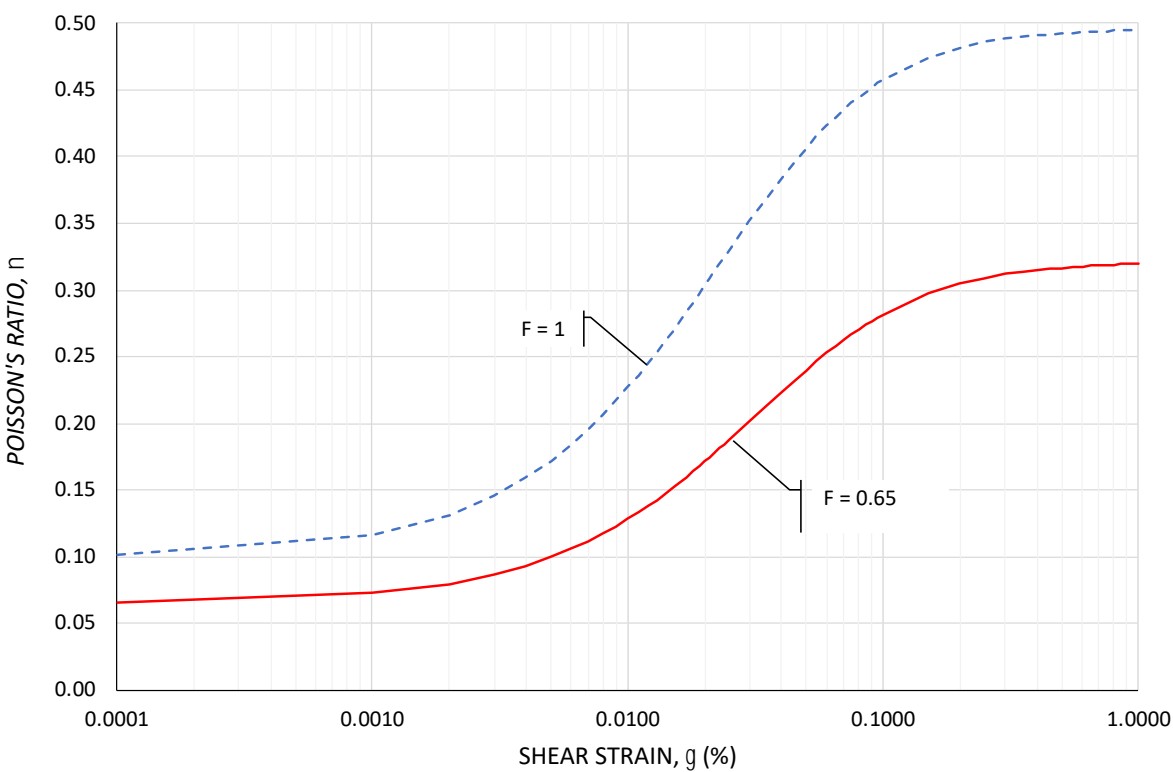

**Figure 11.** Variation in Poisson's ratio as a function of shear strain according to Equation (24) for sand in undrained and drained conditions.

*6.3. Tangent Constrained Modulus*

The following relationship exists between the tangent constrained modulus, $M_t$, and the tangent shear modulus, $G_t$ [49]:

$$M_t = \frac{2(1+v)}{3(1-2v)} G_t \qquad (25)$$

Substituting Equation (21) into Equation (25) yields the general relationship between the tangent constrained modulus, $M_t$, as a function of the small-strain shear modulus, $G_0$ for granular soils. For sandy soil, Equation (22) can be substituted into Equation (25) from which the following relationship is obtained:

$$M_t = G_0 \frac{2(1+v)}{3(1-2v)} \frac{\left(10^{0.5\gamma}\right)\left(7\gamma^2 + 10^{0.5\gamma}\right)}{[14\gamma(1+10^{0.5\gamma})+10^{0.5\gamma}]^2} \qquad (26)$$

According to Equation (9), and assuming for normally consolidated sandy soils that $j = 0.5$, the modulus number, $m$, can be derived from the tangent constrained modulus, $M_t$:

$$m = \frac{M_t}{(\sigma_r \sigma_v')^{0.5}} \qquad (27)$$

Substituting Equation (26) into Equation (27), a relationship for normally consolidated sandy soils between the modulus number, $m$, and the small-strain shear modulus, $G_0$, is obtained,

$$m = \frac{G_0}{(\sigma_r \sigma_v')^{0.5}} \frac{2(1+v)}{3(1-2v)} \frac{\left(10^{0.5\gamma}\right)\left(7\gamma^2 + 10^{0.5\gamma}\right)}{[14\gamma(1+10^{0.5\gamma})+10^{0.5\gamma}]^2} \qquad (28)$$

which is shown in Figure 12. Three different types of sandy soil, according to Table 4 (low, medium, and high), were chosen.

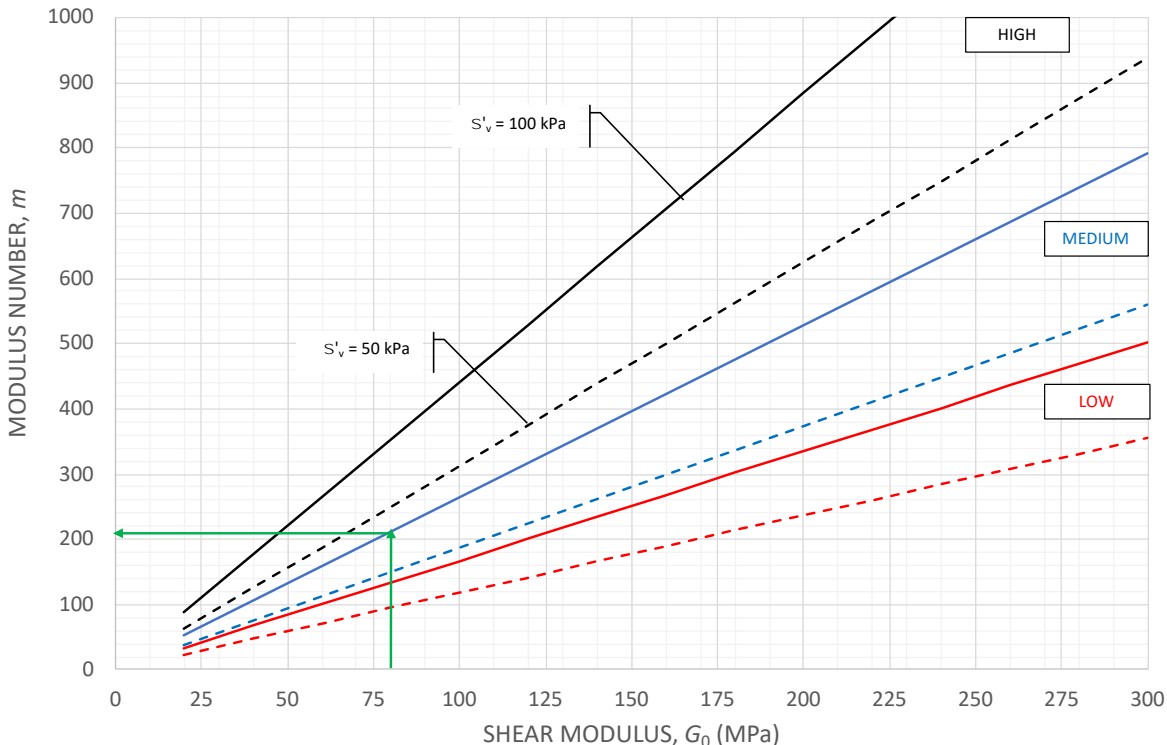

**Figure 12.** Relationship between maximum shear modulus, $G_0$, and modulus number, $m$. Dashed lines $\sigma'_v = 50$ kPa; full line: $\sigma'_v = 100$ kPa. Green arrows indicate the modulus number for medium-dense, normally consolidated sand.

The $m$-values shown in Figure 12 are in good agreement with those proposed by [40,44,45,47].

The following example illustrates the application of the above-described concept, assuming a shear wave speed $C_s = 200$ m/s for a medium sand with a bulk density $\rho = 2000$ kg/m$^3$, $G_0 = 80$ MPa, cf. Equation (1). According to Figure 12, assuming $G_0 = 80$ MPa and a vertical effective stress $\sigma = 100$ kPa, the corresponding modulus number is $m\sim200$ (ranging between 150 and 210). This $m$-value is in good agreement with values for compact sand, cf. Table 5 and Figure 7.

## 7. Discussion

An important challenge in geotechnical engineering is the settlement design of structures founded on granular soils. Empirical and analytical methods can be used to calculate total and differential settlements. The tangent modulus method is a powerful—but not widely used—concept for estimating settlement. The main limitation of all settlement analyses is the selection of realistic input parameters, and particularly the estimation of the constrained modulus, $M_t$. In granular soils, it is difficult to obtain undisturbed soil samples, which can be tested in the laboratory. Therefore, the constrained modulus is frequently estimated based on simplified empirical relationships. Alternatively, correlations between the constrained modulus and results of different types of in situ tests are used. The aim of this investigation was to develop concepts that can be used for the estimation of the constrained modulus, $M_t$, from seismic and static penetration tests.

Seismic field and laboratory tests are used increasingly in geotechnical practice. However, the execution of seismic tests must be well planned. Data interpretation requires experience. The shear wave speed, $C_s$, can be measured in situ with high accuracy, from which the small-strain secant shear modulus, $G_0$, can be derived. A main limitation has been the use of $G_0$ from small-strain tests for settlement analyses, which occur at large strain. Thus, $G_0$ must be related to an equivalent large-strain tangent constrained modulus, $M_t$. The correlation between $G$ and $M$ depends on Poisson's ratio, $\nu$. The data from a

limited number of laboratory tests suggests that Poisson's ratio, $v$, is strain-dependent. This important aspect is frequently neglected.

It has been shown that the loading rate during a seismic test at very low shear strain ($<10^{-4}$%) is surprisingly slow and comparable to that of a conventional static laboratory test. Therefore, the effect of the loading rate on the shear modulus can, for practical purposes, be neglected.

A review of seismic laboratory tests on a wide range of granular soils shows surprisingly good agreement. The most important parameters affecting modulus degradation are shear strain, plasticity index, particle size, and effective confining stress.

A shear strain degradation relationship is proposed that can be used to adjust the small-strain modulus at $10^{-4}$% to a working strain level at approximately 0.1 to 0.5% shear strain.

Applying the tangent modulus method for settlement analyses requires the determination of the constrained modulus, which depends on the modulus number, $m$. By converting the secant shear modulus at small strain, $G_0$, to the tangent constrained modulus, $M_t$, it is possible to estimate the modulus number, $m$. This relationship offers new possibilities for using seismic tests in settlement analyses. For medium sand ($\alpha = 14$, $\beta = 0.5$) at a shear strain level of $\gamma = 0.25$%, which is used to represent static loading, the simple relationship between $G_t$ and $G_0$ is given by Equation (23). Equation (22) gives the general relationship between the shear modulus at very low strain, $G_0$, and the large-strain tangent shear modulus, $G_s$. It is apparent that at a strain level reflecting static loading, the tangent shear modulus is only a fraction of the small-strain shear modulus.

On many ground improvement projects, cone penetration tests are used for design purposes. The compaction specifications are frequently chosen with respect to settlement requirements. In such cases, determining the constrained modulus, based on a stress-adjusted cone resistance, makes it possible to estimate settlements prior to ground treatment and to prescribe the increase in cone resistance that meets the settlement specifications. Rather than using empirical values to correlate cone resistance to an elastic modulus, which is common even in large projects, it is preferable to use the tangent modulus concept. A critical step in the settlement analysis is to adjust the cone resistance with respect to the mean effective stress, as outlined above.

## 8. Conclusions

A review of results from seismic laboratory tests has shown that the most important parameters that affect the shear modulus are shear strain and confining stress. A relatively simple relationship can be used to estimate the degradation of the secant shear modulus. Empirically developed modulus degradation parameters, $\alpha$ and $\beta$, cover a wide strain range ($10^{-4}$ to $10^{-1}$%) and are sufficiently accurate for preliminary design but should be verified for the design of important projects.

A novel relationship is proposed that correlates the small-strain secant shear modulus, $G_0$, to the constrained modulus, $M_t$. For normally consolidated sandy soils, a relationship between the modulus number, $m$, and the small-strain shear modulus, $G_0$, is proposed, which can be used to derive static compression parameters (modulus number) directly from seismic tests. The effect of pre-loading on the modulus number can be taken into account.

To derive the constrained modulus, $M$, from the shear modulus, $G$, it is necessary to estimate the strain-dependent Poisson's ratio, $v$. A relationship is proposed from which Poisson's ratio can be estimated as a function of shear strain.

A relationship between the small-strain shear modulus and the tangent constrained modulus has been derived. The modulus number, $m$, can be calculated from the small strain shear modulus $G_0$.

The modulus number, $m$, can also be estimated based on cone penetration tests. However, this requires that the measured cone resistance be adjusted with respect to the mean effective stress and soil type.

The findings presented in this paper have been applied to a case history where the tangent constrained modulus was measured using different in situ methods. The results have been published in a companion paper.

**Funding:** The author declares that no funding has been received.

**Data Availability Statement:** The data presented in this study are available in the article.

**Acknowledgments:** This paper is dedicated to the late B. B. Broms, my academic advisor and friend over many years. The work by N. Janbu has had a profound influence on the understanding of different factors that govern settlements in soils. The author wishes to extend his sincere gratitude to B.O. Hardin and V.P. Drnevich for the many valuable discussions during his stay at the University of Kentucky. J.C. Santamarina has offered many astute comments regarding the behavior of soils at small-strain loading. The incentive to prepare this paper has been the result of inspiration from the work of J. Burland and H. Poulos. Their influence on the application of small-strain stiffness has been important for the authors. The fruitful discussions with C. Wersäll and B. Fellenius, who have co-authored many papers on this subject, are acknowledged. The valuable comments by the reviewers of the manuscript have improved the quality of the paper; their generous contributions are gratefully acknowledged.

**Conflicts of Interest:** The author declares no conflict of interest.

### Notation List

| | | |
|---|---|---|
| $a$ | = | Empirical modulus factor |
| $C_P$ | = | Compression wave speed |
| CPTU | = | Cone penetration test with pore water pressure measurement |
| $C_{0s}$ | = | Shear wave speed at small strain |
| $C_s$ | = | Strain-dependent shear wave speed |
| $C_U$ | = | Uniformity coefficient ($d_{60}/d_{10}$) |
| $d_{50}$ | = | Particle size at which 50% of the soil is finer |
| $D_{60}$ | = | Particle size at which 60% of the soil is finer |
| DMT | = | Flat dilatometer test |
| $e$ | = | Void ratio |
| $e_0$ | = | Initial void ratio |
| $E$ | = | Young's modulus |
| $E_D$ | = | Dilatometer modulus |
| $E_f$ | = | Young's modulus at failure |
| $E_s$ | = | Young's secant modulus |
| $E_t$ | = | Young's tangent modulus |
| $f$ | = | Frequency |
| $F$ | = | Scaling factor |
| $G$ | = | Shear modulus |
| $G_0$ | = | Shear modulus at small strains |
| $G_s$ | = | Secant shear modulus |
| $G_t$ | = | Tangent shear modulus |
| $I_D$ | = | Material index (DMT) |
| $j$ | = | Stress exponent |
| $k$ | = | Empirical constant, which depends on PI |
| $m$ | = | Modulus number |
| $M$ | = | Constrained modulus |
| $m_1$ | = | Empirically determined parameter |
| $m_2$ | = | Empirically determined parameter |
| $m_r$ | = | Re-loading modulus number |
| $M_s$ | = | Secant constrained modulus |
| $M_t$ | = | Tangent modulus |
| $n$ | = | Porosity |
| $n_1$ | = | Empirically determined parameter |
| $n_2$ | = | Empirically determined parameter |

| $OCR$ | = | Overconsolidation ratio |
|---|---|---|
| $PI$ | = | Plasticity index |
| $q_c$ | = | Cone resistance |
| $q_{cM}$ | = | Stress-adjusted cone resistance |
| RC | = | Resonant column |
| $R_G$ | = | Modulus degradation factor |
| SCPT | = | Seismic down-hole cone penetration test |
| SPLT | = | Screw plate test |
| SPT | = | Standard Penetration Test |
| TS | = | Torsional shear test |
| $S_r$ | = | Degree of saturation |
| $\alpha$ | = | Empirical parameter |
| $\beta$ | = | Empirical parameter |
| $\varepsilon$ | = | Axial strain |
| $\varepsilon_{t,v}$ | = | Volumetric threshold strain |
| $d\varepsilon$ | = | Change in strain |
| $\varepsilon_2$ | = | Strain in horizontal direction |
| $\varepsilon_1$ | = | Strain in vertical direction |
| $\gamma_1$ | = | Shear strain level defining modulus degradation, level 1 |
| $\gamma_2$ | = | Shear strain level defining modulus degradation, level 2 |
| $\gamma_r$ | = | Reference strain |
| $\gamma_{tl}$ | = | Linear threshold strain |
| $\rho$ | = | Bulk density |
| $v'$ | = | Poisson's ratio |
| $\sigma'_0$ | = | Mean effective stress |
| $\sigma_{vo}$ | = | Vertical total stress |
| $\sigma'_p$ | = | Preloading stress |
| $\sigma'_v$ | = | Vertical effective stress |
| $\sigma'_{v1}$ | = | Vertical effective stress after loading |
| $\sigma_r$ | = | Reference stress (equal to 100 kPa) |
| $d\sigma$ | = | Change in stress |
| $\sigma_3$ | = | Horizontal stress |
| $\sigma_1$ | = | Vertical stress |
| $\tau$ | = | Shear stress |

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
