# Peer review of "Determination of Constrained Modulus of Granular Soil from In Situ Tests—Part 1 Analyses"

_geotechnics, doi:10.3390/geotechnics4010002_

Round 1

Reviewer 1 Report

Comments and Suggestions for Authors This research addresses the interrelationship between the small strain shear modulus of soil and corresponding geotechnical parameters, alongside the methodology for deriving the constrained shear modulus using seismic testing data. However, there was a lack of emphasis on the novelty of the study, which was not effectively presented. The majority of the manuscript appears to focus on existing study results. Furthermore, the proposed concept requires further validation. The manuscript exhibits several deficiencies in writing quality, including a disorganized structure and failure to adhere to academic standards for research articles. Here are my detained comments: 

1.      The manuscript title includes "In Situ Tests -1." without an apparent explanation. 

2.      In the abstract, there is an absence of the essential components of an abstract, such as Introduction/Purpose, Methods, Results, and Conclusions/Significance of the research. It primarily revolves around a description of the proposed concept, thus lacking comprehensive coverage. 

3.      Regarding the manuscript's structure, the logical coherence between and within chapters is unclear, thus impeding ease of comprehension. The subdivision within section 6, "Relationship between Shear Modulus and Constrained Modulus," lacks a clear rationale, as each subsection presents a concept in isolation, rather than analyzing their interconnectedness. 

4.      Additionally, the writing quality is insufficient, as evident by four citation errors (line 160, 165, 182, and 426) and an incorrect subtitle in line 155. There is also a statement in line 399-400 ("The tangent shear modulus, Gt, should not be confused with the secant shear modulus, Gs") that appears more suitable for an operational guideline or specification rather than an academic paper.

Author Response

Thank you for the effort and valuable comments. I have tried to include all suggestions in the final manuscript.

Reviewer 2 Report

Comments and Suggestions for Authors

The paper is essentially a review on the topic, which is very useful and contains numerous compilations of data. But the title should emphasize that this is a review. 

It also seems that this is only the 1st part of a series of papers. 

I reccomend to change the title. 

It is not clearly presented what is the proposed method and if this is based on own tests or only on the litterature data. For some graphs is affirmed that the results are in good correlation with the practice, but no data are provided for this. Also, not clear always to which laboratory results the author is referring when affirming that they are in good correlation. 

It is necessary to clarify these points.

Editing is required for the references to figures.

Author Response

(The authors gave the same response as above.)

Reviewer 3 Report

Comments and Suggestions for Authors

The submitted manuscript covers an interesting subject on driving technical parameters from in-situ test records, which is highly praised. Moreover, it is organized and presented favorably and acceptably. However, a few notes are needed to be realized.

1- Since the manuscript is on the application of in-situ tests records, if possible, presenting a typical profile of in-situ tests records along with relevant parameters is highly recommended. Moreover, more details on in-situ records that are used are beneficial.

2-In-situ tests can be categorized as their main output into stiffness and strength tests. Is this issue considered and differentiated in the current study or not?

3-The strain level is among the most important factors in developing models and correlations for designating soil parameters. In other words, the domain and magnitude of parameters may vary in small, medium, and large strain conditions. The given references may be beneficial:

"Sabatini PJ, Bachus RC, Mayne PW, Schneider JA, Zettler TE. Geotechnical Engineering Circular No. 5: Evaluation of Soil and Rock Properties. No. FHWA‐IF‐02‐034; 2002."

"Mayne PW. Stress‐strain‐strength‐flow parameters from enhanced insitu tests. Proceedings of International Conference on In‐Situ Measurement of Soil Properties and Case Histories, Bali, Indonesia. Georgia Institute of Technology; 2001:27‐48."

4-The application and incorporation of CPT records in estimating Janbu parameters (m & j) have been studied and presented previously, which needs to be reviewed somehow. As an example:

"Valikhah, F. & Eslami, A. (2019). CPT-Based Nonlinear StresseStrain Approach for Evaluating Foundation Settlement: Analytical and Numerical Analysis. Arabian Journal for Science and Engineering"

5-In case the recommended issues can not be covered in the main body, they can be addressed in a new section named discussion or remarkable points based on the convenience of the author.

Author Response

(The authors gave the same response as above.)

Round 2

Reviewer 1 Report

Comments and Suggestions for Authors

I have completed the second review of the revised manuscript mentioned above. I believe that the authors have made comprehensive revisions in line with the comments provided during the initial review. They have also clarified their main innovative contribution and central thesis effectively.However, there are still a few areas that require attention:

1. The subtitle of Section 6 remains ambiguous and does not convey the content of the section effectively. I advise the authors to clarify the work performed in this portion by articulating it in the subtitle, rather than presenting a mere label.

2. There appears to be an ongoing issue with the heading numbering at Line 155. This needs to be corrected for a consistent and professional format throughout the document.

3. Regarding my previous comment on the text in Line 421 about the tangent shear modulus (Gt) and the secant shear modulus (Gs), I would like to clarify my position. It's not that the distinction between Gt and Gs is unimportant, but the current phrasing is abrupt and assumes a reader's unfamiliarity with fundamental concepts, which may not be necessary for the intended academic audience. This statement might be better suited as a footnote or in an appendix catering to less familiar readers.

In summary, should these issues be addressed with due diligence, I believe the paper will be strengthened and made suitable for publication. I look forward to seeing these final adjustments.

Author Response

I do appreciate the diligent work of the reviewer. The comments have helped to improve the quality of the manuscript!
